Bone mesenchymal stem cell-derived exosomes prevent hyperoxia-induced apoptosis of primary type II alveolar epithelial cells in vitro

Yang Wei robertyung@163.com 1
Huang Chao 2
Wang Wenjian 3
Zhang Baozhu 4
Chen Yunbin 5
Xie Xinlin 1
1 Department of Pediatrics, The Second Affiliated Hospital of Shenzhen University (The People’s Hospital of Baoan Shenzhen) , Shenzhen , China
2 Department of Traditional Chinese Medicine, The Second Affiliated Hospital of Shenzhen University (The People’s Hospital of Baoan Shenzhen) , Shenzhen , China
3 Department of Respiratory Medicine, Shenzhen Children’s Hospital , Shenzhen , China
4 Department of Oncology, The Second Affiliated Hospital of Shenzhen University (The People’s Hospital of Baoan Shenzhen) , Shenzhen , China
5 Department of Pediatrics, Guangdong Women’s and Children’s Hospital , Guangzhou , China
Sistla Srinivas
Electronic publication date: 2022 Sep 2
Publication date: 2022
Volume: 10
Electronic Location ID: e13692
Received 2022 Mar 28; Accepted 2022 Jun 16
Copyright: ©2022 Yang et al.
Copyright year: 2022
Copyright holder: Yang et al.
License: This is an open access article distributed under the terms of the Creative Commons Attribution License, which permits unrestricted use, distribution, reproduction and adaptation in any medium and for any purpose provided that it is properly attributed. For attribution, the original author(s), title, publication source (PeerJ) and either DOI or URL of the article must be cited.
License URL: https://creativecommons.org/licenses/by/4.0/

Keywords: Hyperoxia, Exosome, AECIIs, LY294002, Rapamycin, PI3K/Akt/mTOR/Ki67

Funding: Science and Technology Planning Project of Shenzhen Municipality # JCYJ20210324111200001 This work was funded by the Science and Technology Planning Project of Shenzhen Municipality (grant # JCYJ20210324111200001). The funders had no role in study design, data collection and analysis, decision to publish, or preparation of the manuscript.

==============================
Background

The presence of alveolar epithelial type II cells (AECIIs) is one of the most important causes of bronchopulmonary dysplasia (BPD). Exosomes from bone mesenchymal stem cells (BMSCs) can reduce hyperoxia-induced damage and provide better results in terms of alveolar and pulmonary vascularization parameters than BMSCs. Currently, intervention studies using BMSC-derived exosomes on the signaling pathways regulating proliferation and apoptosis of alveolar epithelial cells under the condition of BPD have not been reported. This study investigated the effects of rat BMSC-derived exosomes on the proliferation and apoptosis of hyperoxia-induced primary AECIIs in vitro.

Methods

The isolated AECIIs were grouped as follows: normal control (21% oxygen), hyperoxia (85% oxygen), hyperoxia+exosome (20 µg/mL), hyperoxia+exosome+LY294002 (PI3K/Akt inhibitor, 20 µM), and hyperoxia+exosome+rapamycin (mTOR inhibitor, 5 nM). We used the PI3K/Akt inhibitor LY294002 and the mTOR inhibitor rapamycin to determine the roles of the PI3K/Akt and mTOR signaling pathways. The effects of BMSC-derived exosomes on AECII proliferation and apoptosis were assessed, respectively.

Results

Decreased levels of the antiapoptotic protein Bcl-2, the cell proliferation protein Ki67, p-PI3K, p-Akt, and p-mTOR, as well as increased levels of AECII apoptosis and the proapoptotic protein Bax in the hyperoxia group were observed. Notably, Sprague Dawley rat BMSC-derived exosomes could reverse the effect of hyperoxia on AECII proliferation. However, the application of LY294002 and rapamycin inhibited the protective effects of BMSC-derived exosomes.

Conclusion

Our findings revealed that BMSC-derived exosomes could regulate the expression of apoptosis-related proteins likely via the PI3K/Akt/mTOR signaling pathway, thereby preventing hyperoxia-induced AECII apoptosis.

Introduction

Oxygen therapy is an effective therapeutic method for respiratory diseases (Wijsenbeek et al., 2019). However, high oxygen concentrations can induce chronic multisystem diseases such as nerve injury, retinopathy, or bronchopulmonary dysplasia (BPD) in preterm infants (Bui et al., 2017).

BPD is the most common chronic lung disease of preterm infants with immature lung development, thus threatening their survival and quality of life (Anderson & Hillman, 2019; Bonadies et al., 2020). Pathological changes include alveolar dysplasia, decreased numbers of alveoli, and an abnormal distribution of alveolar capillaries. BPD has an incidence rate of 12.3–30% and is prevalent among infants with a gestational age of 22–28 weeks, reaching an incidence rate as high as 68% in this group (Thébaud et al., 2019; Tracy & Berkelhamer, 2019). It poses difficulties in early weaning and an increased incidence of ventilator-associated pneumonia. Long-term oxygen dependency leads to lung oxidation and antioxidant imbalance, inducing inflammation and infection that further aggravate BPD, thus creating a vicious cycle that increases the risk of death (Kalikkot Thekkeveedu, Guaman & Shivanna, 2017). Infants and toddlers with BPD are prone to severe respiratory infections that affect lung function during childhood, adolescence, and even adulthood. Therefore, BPD prevention and treatment are key focus areas in neonatal research, particularly in the areas of pulmonary epithelial cell damage and diffuse alveolar injury (Jensen & Schmidt, 2014).

In animal models, oxygen alone or positive pressure ventilation can lead to BPD-associated changes, alveolar simplification, and microvascular injury (Jobe, 2015). Hyperoxia-induced BPD can arise due to the abnormal repair of lung tissues, thereby inhibiting the apoptosis and proliferation of alveolar epithelial cells (Endesfelder et al., 2020; Saugstad, 2003). The regenerative capabilities of type II alveolar epithelial cells (AECIIs) underlie recovery after alveolar epithelial injury (Hou et al., 2015; Wu et al., 2018). Thus, aberrant proliferation and differentiation of AECIIs in damaged lung tissues affect normal repair and recovery of the alveolar structure and function (Wu et al., 2018; Yang et al., 2014).

Exosomes are the smallest identified extracellular vesicles that contain proteins, enzymes, or growth factors, with diameters ranging between 40 and 150 nm. They can be released by several cell types in vitro and have been detected in biological fluids in vivo (Braun et al., 2018; Lesage & Thébaud, 2018). Exosomes mediate intercellular communication and have potential applications in clinical therapy (Collins, 2020; Gortner et al., 2012; Li et al., 2020b). Of note, treatment with exosomes has shown significant reductions in oxidative damage and apoptosis of AECIIs (Braun et al., 2018).

Given the known role of PI3K/Akt/mTOR/Ki67 signaling in regulating proliferation and apoptosis (Feng et al., 2019; Rahmani et al., 2020; Samakova et al., 2019; Zhang et al., 2020a; Zhang et al., 2021; Zhang et al., 2020b; Zhang et al., 2020c), in this study, we investigated its role in exosome-mediated regulation of AECII proliferation and apoptosis during hyperoxia-induced BPD using hyperoxia rat models. We examined the effect of bone mesenchymal stem cell (BMSC)-derived exosomes on the proliferation and apoptosis of hyperoxia-induced primary AECIIs using a cell viability assay, annexin V-FITC staining, and western blot analysis. Furthermore, we used PI3K/Akt and mTOR inhibitors to confirm the protective effects of exosomes on AECIIs under hyperoxia conditions. Our findings provide preliminary insights into the underlying mechanism of BMSC-derived exosomes on hyperoxia-induced AECII proliferation. The knowledge gained from this study could have potential clinical therapeutic applications in the future.

Materials & Methods

BMSC isolation and culture

Animal experiments were performed with approvals from the Laboratory Animal Welfare Ethics Committee of the Medicine Center at Shenzhen Peking University and Hong Kong University of Science and Technology, China (approval number: 2021-147). For the experiments, sixteen 6-week-old Sprague Dawley rats weighing about 200 g (Experimental Animal Center of Southern Medical University, Guangzhou, China) were housed in steel cages at 18–22 °C under 50–60% relative humidity. A standard pellet diet and water were provided ad libitum. The rats were sacrificed by cervical dislocation after 1 week, and the femur and tibia were collected under aseptic conditions. The bone marrow cavity was washed with Dulbecco’s modified Eagle medium (DMEM; Gibco, Lifetech, China), filtered and centrifuged at 1,000 rpm for 5 min. Next, the BMSCs were resuspended in DMEM containing 10% fetal bovine serum (FBS; Gibco, Thermo Fisher Scientific, Waltham, MA, USA), seeded (1 × 106/mL) in 10-cm culture plates, and cultured for 3 days before refreshing half of the medium with fresh culture medium. At approximately 80–90% confluency (after ∼10 days of cell culture), the BMSCs were trypsinized (0.25% trypsin, Gibco, Thermo Fisher Scientific, Waltham, MA, USA). BMSCs from passages 4 to 8 were used for subsequent experiments. The osteogenic and adipogenic potentials of the stem cells were determined by Alizarin red staining and oil red staining, respectively, as described previously (Futrega et al., 2018). The same differentiation medium and staining solution were used for skin fibroblasts as the negative control group.

Isolation of AECIIs from pregnant rats and culture grouping

The AECIIs were isolated from six 4-week-old pregnant Sprague Dawley rats weighing approximately 80–100 g (Experimental Animal Center of Southern Medical University, Guangzhou, China). Primary AECIIs were isolated as previously described (Gonzalez & Dobbs, 2013), with modifications. Briefly, pregnant rats at 20 days of gestation were first anesthetized intraperitoneally with 3% pentobarbital sodium (50 mg/kg; Shanghai Sangong Bioengineering, Shanghai, China) and then subjected to uterine-incision delivery to induce premature labor. Fetal lungs were removed from the thoracic cavity, rinsed with precooled phosphate-buffered saline (PBS), diced into small pieces, and trypsinized(0.25% trypsin) for 25 min at 37 °C. DMEM containing 10% FBS was added to the cell suspension to stop the digestion, and then the cells were filtered and centrifuged at 1,000 rpm for 5 min. The cell pellets obtained were incubated with collagenase IV (10 µg/mL) for 15 min at 37 °C. DMEM containing 10% FBS was then added to stop the reaction. Next, the cell suspension was centrifuged, resuspended in DMEM, and then inoculated in a 10-cm culture plate. After the second passage, the primary AECIIs were incubated with the AECII-specific surface marker surfactant protein C (SP-C) antibody followed by the corresponding FITC-conjugated secondary antibody for detection by confocal microscopy (Leica). We also detected lamellar bodies in ACEIIs using a JEOL 1400 transmission electron microscope. The isolated AECIIs were divided into five culture conditions: normal control (NC; 21% oxygen), hyperoxia (85% oxygen), hyperoxia+exosome (20 µg/mL), hyperoxia+exosome+LY294002 (a PI3K/Akt inhibitor, Sigma, Shanghai, China; 20 µM), and hyperoxia+exosome+rapamycin (an mTOR inhibitor, Sigma, Shanghai, China; 5 nM).

Isolation and characterization of BMSC-derived exosomes

BMSC-derived exosomes cultured at 37 °C under 5% CO2 for 12 h were harvested from FBS-free culture medium using Total Exosome Isolation Reagent (Life Technology, Shanghai, China), according to the manufacturer’s instructions. The isolated BMSC-derived exosomes were then washed three times with PBS. As described previously (Zhang & Chan, 2010), the BMSC-derived exosomes were deposited on Formvar/carbon-coated electron microscopy grids and allowed to air dry. Subsequently, the BMSC-derived exosomes were fixed with 2% glutaraldehyde, counterstained with 4% uranyl acetate, and then scanned using a JEOL 1400 transmission electron microscope (JEOL Ltd., Tokyo, Japan). The BMSC-derived exosomes were assessed by detection of two well-characterized exosome surface protein markers, CD9 and CD63, through nanoparticle tracking analysis with a ZetaView® Nanoparticle Tracking Analyzer (Particle Metrix, Meerbusch, Germany) and western blot analysis.

Cell counting kit-8 (CCK8) cell viability assay

AECIIs (5 × 103 cells/well) were seeded in 96-well plates and cultured in DMEM containing 10% FBS for 6 h under 85% oxygen before exosomes and/or LY294002 or rapamycin inhibitor were added. After culturing for 24, 48, and 72 h, respectively, the AECII cell viability was assessed at 490 nm by the CCK8 assay (Beyotime Biotech, Shanghai, China), according to the manufacturer’s instructions. Briefly, AECII cultures were incubated with 0.5 mg/mL CCK8 for 20 min before measurements were taken.

Annexin V-FITC apoptosis assay

AECII apoptosis was assessed using an annexin V-FITC apoptosis detection kit (Beyotime Biotech, Jiangsu, China). After treatment with exosomes and/or LY294002 or rapamycin inhibitor for 24 h, AECIIs were spun down and incubated for 15 min at room temperature in the dark. Next, the cell pellets were resuspended in 500 µL of binding buffer containing 5 µL of annexin V-FITC and 5 µL of propidium iodide. The AECIIs were prepared using a flow cytometry assay kit (Abcam, Cambridge, UK) and were then counted using a BD Accuri C6 Plus flow cytometer (BD Biosciences, Franklin Lakes, NJ, USA).

Western blotting

Total protein was extracted from AECIIs treated with BMSC-derived exosomes and/or inhibitors using radioimmunoprecipitation assay lysis buffer (Beyotime Biotech, Shanghai, China). The protein concentration was determined using a bicinchoninic acid kit (Beyotime Biosciences, Shanghai, China). Proteins were separated by gel electrophoresis, and the bands were transferred onto polyvinylidene difluoride membranes (Millipore, Burlington, MA, USA). The membranes were first incubated with mouse anti-human AKT, p-AKT, p-mTOR, mTOR, Bax, Bcl-2, or Ki67 primary antibodies (Santa Cruz, USA) at 4 °C overnight, followed by the corresponding anti-rabbit and anti-mouse secondary antibodies (BOSTER Technology, Wuhan, China) at room temperature for 1 h. The bands were visualized using the enhanced chemiluminescence substrate and were then analyzed using ImageJ software (NIH, Bethesda, MD, USA). GAPDH (Sanjian Biology, Tianjin, China) was used as the loading control.

Statistical analysis

Statistical analysis was performed using SPSS v20 software. Data from three independent experiments were expressed as the mean  ± standard deviation (SD). Comparison among groups was performed by one-way analysis of variance, while the two-tailed Student’s t-test was used to compare groups. Values of P < 0.05 were considered statistically significant.

Results

Morphology and stem cell characteristics of BMSCs

Unlike the BMSCs seeded at day1 (Fig. 1A), the cells appeared as fibroblast-like mononuclear cells and underwent logarithmic growth in the second passage (Fig. 1B). The adipogenic and osteogenic potential of the BMSCs were confirmed by oil red staining (Figs. 1C and 1D) and Alizarin red staining (Figs. 1E and 1F), respectively, indicating successful isolation of rat BMSCs. Fibroblasts were not positively stained by either oil red or Alizarin red (Figs. 1D and 1F).

Figure 1 Light microscopy images of BMSC morphology and stem cell characteristics.

(A) BMSC suspension at day 1 of culture. (B) Adherent BMSCs at day 12 of culture. (C) Oil red staining of BMSCs demonstrate their adipogenicity. (D) Oil red staining of fibroblasts as a negative control. (E) Alizarin red staining of BMSCs demonstrate their osteogenecity. (F) Alizarin red staining of fibroblasts as a negative control.

Characterization of BMSC-derived exosomes

From transmission electron microscopy,we observed particles with diameters of 40–100 nm, which is characteristic of exosomes (Fig. 2A). Vesicles with an average particle diameter size of 100–200 nm and a maximum diameter of 600 nm were detected from nanoparticle tracking analysis (Fig. 2B). Western blot analysis also confirmed successful isolation of BMSC-derived exosomes (Fig. 2C).

Figure 2 Identification of BMSC-derived exosomes and lamellar bodies of AECIIs.

(A) Transmission electron microscopy image showing exosomes (scale bar = 100 nm). (B) Profile showing the diameters of isolated BMSC-derived exosomes from BMSCs. (C) Western blot detection of exosomes by their characteristic CD9 and CD63 markers. (D) Transmission electron microscopy image of lamellar bodies (scale bar = 200 nm).

Morphology and characterization of ACEIIs

Typical lamellar bodies were detected by transmission electron microscopy (Fig. 2D). AECIIs formed colonies and exhibited a cuboidal cobblestone morphology under a light microscope (Fig. 3A). Detection of the commonly expressed AECII marker SP-C by immunofluorescence also confirmed successful AECII isolation (Figs. 3B, 3C and 3D).

Figure 3 Isolation, identification, and activity of rat AECIIs in the five experimental groups: normal control (NC), hyperoxia, hyperoxia+exosome, hyperoxia+exosome+LY294002, and hyperoxia+exosome+rapamycin.

(A) Representative confocal microscopy image of adherent AECIIs. Representative immunofluorescence images of (B) DAPI-stained AECIIs, (C) surfactant protein C (SP-C)/FITC-labeled AECIIs, and (D) merged images of B and C. (E) CCK8 cell viability assay profile of AECIIs. # P < 0.05, * P < 0.01, ** P < 0.001, ns, not statistically significant.

BMSC-derived exosomes protect AECIIs from hyperoxia-induced apoptosis

Next, we determined the effect of exosomes on AECII cell viability and apoptosis under hyperoxia. The CCK8 assay showed no significant changes in cell viability for all groups at 24 h, but the cell viability was significantly decreased in the hyperoxia group at 48 h (t = 6.3246, P = 0.0032) and 72 h (t = 16.5132, P = 0.0001) compared with the normal control group (Fig. 3E). Of note, hyperoxia-mediated growth inhibition was significantly reduced in the hyperoxia+exosome group at 48 h (t = 7.8558, P = 0.0014) and 72 h (t = 10.4151, P = 0.0005), compared with the hyperoxia group (Fig. 3E). The hyperoxia+exosome+LY294002 and hyperoxia+exosome+rapamycin groups showed significantly lower cell viabilities, compared with the hyperoxia+exosome group at 48 h (Fig. 3E; hyperoxia+exosome+LY294002 vs.hyperoxia+exosome, t = 8.8822, P = 0.0009; hyperoxia+exosome+rapamycin vs.hyperoxia+exosome, t = 9.8590, P = 0.0006) and 72 h (Fig. 3E, hyperoxia+exosome+LY294002 vs.hyperoxia+exosome, t = 11.4614, P = 0.0003; hyperoxia+exosome+rapamycin vs.hyperoxia+exosome, t = 9.3915, P = 0.0007).

There were significant differences of early and late apoptosis or necrosis of AECIIs between the normal control group (Fig. 4A) and the hyperoxia group (Fig. 4B, t = 27.3758, P = 0.0000). In addition, a marked decrease in apoptotic and necrotic AECIIs was observed for the hyperoxia+exosome group (Fig. 4C, t = 22.8889, P = 0.0000); however, the effects were reversed for the hyperoxia+exosome+LY294002 group (Fig. 4D, t = 15.6880, P = 0.0001) and the hyperoxia+exosome+rapamycin group (Fig. 4E, t = 27.0877, P = 0.0000).

Figure 4 Flow cytometry and semi-quantitative profiles of AECII apoptosis.

(A–E) Flow cytometry profiles of AECIIs of the five experimental groups. (F) Plot of the percentage of AECIIs in the five experimental groups. Data are shown as the mean ± SD; n = 3. ** P < 0.05 vs. hyperoxia; # P < 0.01 vs. hyperoxia.

Exosomes affect the levels of apoptosis-related proteins and the phosphorylation of key factors of the PI3K/Akt/mTOR signaling pathway

The effect of BMSC-derived exosomes on hyperoxia-induced apoptosis of AECIIs was assessed by western blot analysis (Figs. 5A and 5B). The Bax/Bcl-2level was significantly increased, while the Ki67 level was decreased in the hyperoxia group, compared with the normal control group (Figs. 5A and 5B; t = 6.1929, P = 0.0035; t = 3.3567, P = 0.0284, respectively). In addition, Akt, PI3K, and mTOR phosphorylation was inhibited in the hyperoxia group but was rescued in the hyperoxia+exosome group (Figs. 5A and 5B; t = 25.5119, P = 0.0000; t = 9.2418, P = 0.0008; t = 9.0355, P = 0.0008, respectively). Furthermore, decreased p-Akt and Ki67 protein levels as well as mTOR phosphorylation were noted in the hyperoxia+exosome+LY294002 group, compared with the hyperoxia+exosome group (Figs. 5A and 5C; t = 21.4586, P = 0.0000; t = 6.5485, P = 0.0028; t = 14.1400, P = 0.0001, respectively).

Figure 5 Effect of BMSC-derived exosomes on the expression of apoptosis-related proteins in AECIIs.

(A) Western blot analysis of apoptosis-related proteins and key proteins of the PI3K/Akt/mTOR signaling pathway. Histograms of (B) the phosphorylation levels of p-PI3K, p-Akt, and p-mTOR and the Bax/Bcl2 ratio. (C) The Ki67 level in the five experimental groups. Data are shown as the mean ± SD from three independent experiments (n = 3). ** P < 0.01 and * P < 0.05; normal control (NC) vs. hyperoxia. ## P < 0.01 and # P < 0.05; exosome+hyperoxia vs. hyperoxia.

Discussion

Hyperoxia induces the generation of oxidative free radicals that promote AECII apoptosis. In addition, treatment with exosomes can improve lung tissue histology and promote macrophage M2 polarization, thereby reducing inflammation in a hyperoxia-induced BPD mouse model (Willis et al., 2018a). Moreover, exosomes have been reported to increase pulmonary vascularity, promote alveolar formation, and restore interrupted alveolar growth during tissue hyperoxia (Collins, 2020; Lesage & Thébaud, 2018; Willis et al., 2018b; Willis et al., 2020). Exosomes possess the mesenchymal stem cell markers CD29, CD90, CD73, and CD44 as well as the exosome-specific markers CD9, CD63, and CD81 (Klymiuk et al., 2019). Thus, BMSC-derived exosomes possess stem cell-like therapeutic properties and were studied for their potential application in clinical therapy. In this study, we demonstrated the stem cell characteristics of rat BMSCs by confirming the adipogenetic and osteogenic potential using oil red O and Alizarin red staining.

In this study, we explored the efficacy of BMSC-derived exosomes in alleviating hyperoxia-induced AECII damage in Sprague Dawley rats. We examined the morphology of BMSC-derived exosomes using transmission electron microscopy and confirmed the successful isolation of BMSC-derived exosomes from rat AECIIs by western blotting. Several signaling pathways like the PI3K/Akt/Foxo3a (Wu et al., 2018) and PI3K/Akt/mTOR (Gu et al., 2020) pathways have been implicated in exosome-mediated inhibition of hyperoxia-induced AECII apoptosis. Recently, it has been reported that BMSC-derived exosomes rich in miR-30B-3p can effectively inhibit AECII apoptosis and promote its proliferation in lipopolysaccharide-induced acute lung injury by downregulating the target gene SAA3 (Yi et al., 2019). It has also been shown that BMSC-derived exosomes can reverse the apoptosis of AECIIs in acute lung injury by inhibiting the NRF-2/ARE and NF- κB signaling pathways (Li et al., 2020a). In this study, we noted decreased Akt and mTOR phosphorylation and Ki67 levels, decreased AECII proliferation, but increased apoptosis in the hyperoxia group. Of note, treatment with exosomes appeared to reverse the effects seen in the hyperoxia rats. However, application of the PI3K/Akt blocker LY294002 and the mTOR inhibitor rapamycin reversed the protective effects of BMSC-derived exosomes on AECII proliferation in rats under hyperoxia. Thus, treatment with exosomes likely reversed the effects in the hyperoxia+exosome group, compared with that for the hyperoxia group.

Figure 6 Schematic illustration of the protective effect of exosomes on AECII proliferation under hyperoxia.

A limitation of this study is that it was conducted in vitro and in-vivo data are lacking. Therefore, we will aim to confirm our findings using animal models in a follow-up study. Elucidation of the critical component of exosomes that confers protective effects to AECIIs via the PI3K/Akt/mTOR/Ki67 pathway will be the focus of future research, which could potentially advance BPD clinical therapy.

Conclusions

Our results suggest the potential role of the PI3K/Akt/mTOR/Ki67 pathway in regulating exosome-mediated inhibition of hyperoxia-induced AECII apoptosis (Fig. 6). In particular, the expression of antiapoptotic proteins was elevated while that of proapoptotic proteins was downregulated in the hyperoxia+exosome group. Despite the promise of stem cell therapy, challenges like cell heterogeneity, tumorigenicity, immune regulation, and secretion of inflammatory factors (Riester et al., 2020) must be overcome to realize the full potential of effective applications of stem cells in clinical treatment.

Supplemental Information

Data S1 Raw data

Click here for additional data file.

Supplemental Information 2 Author Checklist - Full

Click here for additional data file.

Supplemental Information 3 Uncropped Gels/Blots

Click here for additional data file.

Additional Information and Declarations

Competing Interests

Author Contributions

Animal Ethics

Data Availability

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

The authors declare there are no competing interests.

Wei Yang conceived and designed the experiments, performed the experiments, analyzed the data, prepared figures and/or tables, authored or reviewed drafts of the article, and approved the final draft.

Chao Huang conceived and designed the experiments, performed the experiments, analyzed the data, prepared figures and/or tables, authored or reviewed drafts of the article, and approved the final draft.

Wenjian Wang analyzed the data, authored or reviewed drafts of the article, and approved the final draft.

Baozhu Zhang performed the experiments, prepared figures and/or tables, and approved the final draft.

Yunbin Chen conceived and designed the experiments, authored or reviewed drafts of the article, and approved the final draft.

Xinlin Xie performed the experiments, prepared figures and/or tables, and approved the final draft.

The following information was supplied relating to ethical approvals (i.e., approving body and any reference numbers):

Animal experiments were performed with approvals from the Laboratory Animal Welfare Ethics Committee of the Medicine Center at Shenzhen Peking University and Hong Kong University of Science and Technology, China (approval number: 2021-147).

The following information was supplied regarding data availability:

The raw data are available in the Supplemental Files.