# Peer review of "Bone mesenchymal stem cell-derived exosomes prevent hyperoxia-induced apoptosis of primary type II alveolar epithelial cells in vitro"

_PeerJ, doi:10.7717/peerj.13692_

## Round 0.1 · original submission · Major Revisions

Please check the comments made by the reviewers and make suitable edits and resubmit at the earliest.

Reviewer 1 ·

Basic reporting

I have trouble accepting the figures (specifically the immunoblots).

Experimental design

No rationale for using PI3K/Akt inhibitor, 20 μM or mTOR inhibitor, 5 μM. IC 50 studies required for selection of this concentration.

Validity of the findings

Figure 2C is unacceptable in its present form.
Figure 2C. No endogenous control run and reported in the western

Figure 5A is unacceptable in its present form.
An endogenous control (GAPDH) is required per blot run.
Figure 5A shows 3 phospho-proteins and their respective total protein. It is impossible to run all 3 phospho protein on the same gel. 3 GAPDH needs to be accounted for.

In addition there are 3 more protein shown in Fig 5A. One GADH for all proteins is mis-leading.

Additional comments

Manuscript in present format is not acceptable for publication. Authors need to re-do all immunoblots and understand the basics in reporting a western data. Endogenous controls are a requisite per blot.
If, what they claim is true, additional work required on reversal effects.

Reviewer 2 ·

Basic reporting

Minor edits required, but looks good overall

Experimental design

No comment

Validity of the findings

No comment

Reviewer 3 ·

Basic reporting

The manuscript is clear and well written, which is easy to follow. The authors have included appropriate background information and references, although the introduction can be improved with additional details about the hyperoxia-induced BPD and its mechanism. The hypothesis of the experiment is clearly stated.

Experimental design

The figures included are relevant and well presented. However, the labels in Figure 4 are hard to read, which can be improved. The hypothesis for the experiment is well described, robust, and sound and would add to the existing knowledge of ACEIIs. The authors describe the methods section well. Although, there is no information regarding the sex of the rats used for these experiments. Were both males and females used? The western blot section could be improved if the authors could provide the information regarding the dilutions used for the various proteins in the experiment to make it easy to replicate. The discussion section needs improvement. The authors have mostly stated their findings in the discussion section, and I would like to see the thought process of the authors as to why they think that exosomes prevent hyperoxia-induced apoptosis in AECIIs. There is a need for more discussion.

Validity of the findings

No comment.

Additional comments

Overall, this is a strong, novel, and essential study; however, the manuscript needs improvement.

---

## Round 0.2 · accepted · Accept

Ver happy to see that the paper is revised to the reviewer suggestions and consider this paper for publication

Reviewer 3 ·

Basic reporting

No comment

Experimental design

No comment

Validity of the findings

No comment

Additional comments

The authors have addressed all the questions I asked.